# Contrastive Fine-grained Class Clustering via Generative Adversarial Networks

**Yunji Kim**
NAVER AI Lab
yunji.kim@navercorp.com

**Jung-Woo Ha**
NAVER AI Lab & NAVER CLOVA
jungwoo.ha@navercorp.com

## Abstract

Unsupervised fine-grained class clustering is a practical yet challenging task due to the difficulty of feature representations learning of subtle object details. We introduce C3-GAN, a method that leverages the categorical inference power of InfoGAN with contrastive learning. We aim to learn feature representations that encourage a dataset to form distinct cluster boundaries in the embedding space, while also maximizing the mutual information between the latent code and its image observation. Our approach is to train a discriminator, which is also used for inferring clusters, to optimize the contrastive loss, where image-latent pairs that maximize the mutual information are considered as positive pairs and the rest as negative pairs. Specifically, we map the input of a generator, which was sampled from the categorical distribution, to the embedding space of the discriminator and let them act as a cluster centroid. In this way, C3-GAN succeeded in learning a clustering-friendly embedding space where each cluster is distinctively separable. Experimental results show that C3-GAN achieved the state-of-the-art clustering performance on four fine-grained image datasets, while also alleviating the mode collapse phenomenon. Code is available at https://github.com/naver-ai/c3-gan.

## 1 Introduction

Unsupervised fine-grained class clustering is a task that classifies images of very similar objects (Benny & Wolf, 2020). While a line of works based on multiview-based self-supervised learning (SSL) methods (He et al., 2020; Chen et al., 2020; Grill et al., 2020) shows promising results on a conventional coarse-grained class clustering task (Van Gansbeke et al., 2020), it is difficult for a fine-grained class clustering task to benefit from these methods for two reasons. First, though it is more challenging and ambiguous for finding distinctions between fine-grained classes, datasets for this type of task are difficult to be large-scale. We visualized this idea in Figure 1 which shows that finding distinctions between fine-grained classes is more difficult than doing so for coarse-grained object classes. Second, the augmentation processes in these methods consider subtle changes in color or shape, that actually play an important role in differentiating between classes, as noisy factors. Thus, it is required to find an another approach for a fine-grained class clustering task.

Generative adversarial networks (GAN) (Goodfellow et al., 2014) can be a solution as it needs to learn fine details to generate realistic images. A possible starting point could be InfoGAN (Chen et al., 2016), that succeeded in unsupervised categorical inference on MNIST dataset (LeCun et al., 2010) by maximizing the mutual information between the latent code and its observation. The only prior knowledge employed was the number of classes and the fact that the data is uniformly distributed over the classes. FineGAN (Singh et al., 2019) extends InfoGAN by integrating the scene decomposition method into the framework, and learns three latent codes for the hierarchical scene generation. Each code, that is sampled from the independent uniform categorical distribution, is sequentially injected to multiple generators for a background, a super-class object, and a sub-class object image syntheses. FineGAN demonstrated that two latent codes for object image generations could be also utilized for clustering real images into their fine-grained classes, outperforming conventional coarse-grained class clustering methods. This result implies that extracting only foreground features is very helpful for the given task. However, FineGAN requires object bounding box annotations and additional training of classifiers, which greatly hinder its applicability. Also, the method lacks the ability of learning the distribution of real images, due to the mode collapse phenomenon (Higgins et al., 2017).

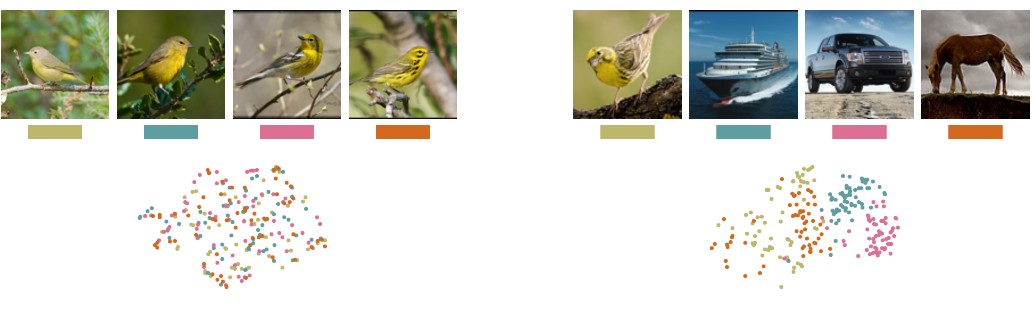

(a) Fine-grained Class Clustering          (b) Coarse-grained Class Clustering

Figure 1: **Fine-grained vs. Coarse-grained.** We compare two sets of 4 classes, each selected from (a) CUB (Wah et al., 2011) and (b) STL (Coates et al., 2011) datasets. The diagrams visualize image features extracted from ResNet-18 that was trained on each dataset via the method of SimCLR (Chen et al., 2020). The color of each dot corresponds to its ground-truth class. Both in the image-level and the feature-level, the clusters in (a) are way more indistinguishable than those in (b).

In this paper, we propose **C**onstrastive fine-grained **C**lass **C**lustering **GAN** (C3-GAN) that deals with the aforementioned problems. We first remove the reliance on human-annotated labels by adopting the method for unsupervised scene decomposition learning that is inspired by PerturbGAN (Bielski & Favaro, 2019). We then adopt contrastive learning method (van den Oord et al., 2018) for training a discriminator by defining pairs of image-latent features that maximize the mutual information of the two as positive pairs and the rest as negative pairs. This is based on the intuition that optimal results would be obtained if cluster centroids are distributed in a way that each cluster can be linearly separable (Wang & Isola, 2020). In specific, we map the input of a generator that was sampled from the categorical distribution onto the embedding space of the discriminator, and let it act as a cluster centroid that pulls features of images that were generated with its specific value. Since the relations with other latent values are set as negative pairs, each cluster would be placed farther away from each other by the nature of the contrastive loss. We have conducted experiments on four fine-grained image datasets including CUB, Stanford Cars, Stanford Dogs and Oxford Flower, and demonstrated the effectiveness of C3-GAN by showing that it achieves the state-of-the-art clustering performance on all datasets. Moreover, the induced embedding space of the discriminator turned out to be also helpful for alleviating the mode collapse issue. We conjecture that this is because the generator is additionally required to be able to synthesize a certain number of object classes with distinct characteristics.

Our main contributions are summarized as follows:

- We propose a novel form of the information-theoretic regularization to learn a clustering-friendly embedding space that leads a dataset to form distinct cluster boundaries without falling into a degenerated solution. With this method, our C3-GAN achieved the state-of-the-art fine-grained class clustering performance on four fine-grained image datasets.

- By adopting scene decomposition learning method that does not require any human-annotated labels, our method can be applied to more diverse datasets.

- Our method of training a discriminator is not only suitable for class clustering task, but also good at alleviating the mode collapse issue of GANs.

## 2   RELATED WORK

**Unsupervised Clustering.** Unsupervised clustering methods can be mainly categorized into the information theory-based and Expectation–Maximization(EM)-based approaches. The first approach aims to maximize the mutual information between original images and their augmentations to train a model in an end-to-end fashion (Ji et al., 2019; Zhong et al., 2020). To enhance the performance, IIC (Ji et al., 2019) additionally trains the auxiliary branch using an unlabeled large dataset, and DRC (Zhong et al., 2020) optimizes the contrastive loss on logit features for reducing the intra-class feature variation. EM-based methods decouple the cluster assignment process (E-step) and the feature representations learning process (M-step) to learn more robust representations. Specifically, the

feature representations are learnt by not only fitting the proto-clustering results estimated in the expectation step, but also optimizing particular pretext tasks. The clusters are either inferred by $k$-means clustering (Xie et al., 2016; Caron et al., 2018; Liu et al., 2020; Li et al., 2021) or training of an additional encoder (Dizaji et al., 2017). However, many of these methods suffer from an uneven allocation issue of $k$-means clustering, which could result in a degenerate solution. For this reason, SCAN (Van Gansbeke et al., 2020) proposes the novel objective function that does not require the $k$-means clustering process. Based on the observation that $K$ nearest neighbors of each feature point belong to the same ground-truth cluster with a high probability, it trains a classifier that assigns identical cluster id to all nearest neighbors, which significantly improved the clustering performance. Unlike these prior methods where the cluster centroids are not learnt in an end-to-end manner or not the subject of interest, C3-GAN investigate and try to improve their distribution for achieving better clustering performance.

**Unsupervised Scene Decomposition.** Heavy cost of annotating segmentation masks have drawn active research efforts on developing unsupervised segmentation methods. Some works address the task in the information-theoretic perspective (Ji et al., 2019; Savarese et al., 2021), but a majority of the works are based on GANs. ReDO (Chen et al. (2019)) learns to infer object masks by redrawing an input scene, and PerturbGAN (Bielski & Favaro (2019)), that has a separate background and foreground generator, triggers scene decomposition by randomly perturbing foreground images. Meanwhile, another line of works utilizes pre-trained high quality generative models such as Style-GAN (Karras et al., 2019; 2020) and BigGAN (Brock et al., 2019). Labels4Free (Abdal et al., 2021) trains a segmentation network on top of the pre-trained StyleGAN, utilizing the fact that inputs of each layer of StyleGAN have different degrees of contribution to foreground synthesis. Voynov & Babenko (2020) and Melas-Kyriazi et al. (2021) explore the latent space of pre-trained GANs to find the perturbing directions that can be used for inducing foreground masks. While all these methods aim to infer foreground masks by training an additional mask predictor with synthesized data, or projecting real images into the latent space of the pre-trained GAN, C3-GAN is rather focusing on the semantic representations learning of foreground objects.

**Fine-grained Feature Learning.** For fine-grained feature representations learning, some works (Singh et al., 2019; Li et al., 2020; Benny & Wolf, 2020) have extended InfoGAN by integrating the scene decomposition learning method into the framework with multiple pairs of adversarial networks. FineGAN (Singh et al., 2019) learns multiple latent codes to use them for sequential generation of a background, a super-class object, and a sub-class object image, respectively. MixNMatch (Li et al., 2020) and OneGAN (Benny & Wolf, 2020) extend FineGAN with multiple encoders to directly infer these latent codes from real images and use them for manipulating images. Since this autoencoder-based structure could evoke a degenerate solution where only one generator is responsible for synthesizing the entire image, MixNMatch conducts adversarial learning on the joint distribution of the latent code and image (Donahue et al., 2017), while OneGAN trains a model in two stages by training generators first and encoders next. Even though these works have succeeded in generating images in a hierarchical manner and learnt latent codes that can be used for clustering real images corresponding to their fine-grained classes, they cannot be applied to datasets that have no object bounding box annotations. Moreover, they require a training of additional classifiers using a set of generated images annotated with their latent values. The proposed model in this paper, C3-GAN, can learn the latent code of foreground region in a completely unsupervised way, and simply utilize the discriminator for inferring the clusters of a given dataset.

## 3 METHOD

Given a dataset $X = \{x_i\}_{i=0}^{N-1}$ consisting of single object images, we aim to distribute data into $Y$ semantically fine-grained classes. Our GAN-based model infers the clusters of data in the semantic feature space $\mathcal{H} \in \mathbb{R}^{d^h}$ of the discriminator $D$. The feature space $\mathcal{H}$ is learnt by maximizing the mutual information between the latent code, which is the input of a generator, and its image observation $\hat{x}$. For more robust feature representations learning, we decompose a scene into a background and foreground region and associate the latent code mainly with the foreground region. We especially reformulate the information-theoretic regularization to optimize the contrastive loss defined for latent-image feature pairs to induce each cluster to be linearly separated in the feature space.

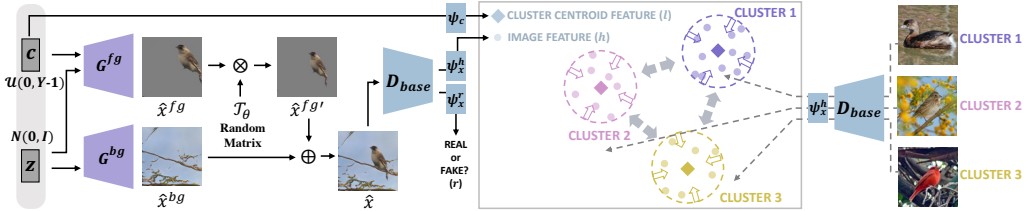

Figure 2: Overview of C3-GAN. It synthesizes a background image $\hat{x}^{bg}$ and a foreground image $\hat{x}^{fg}$ from the background generator $G^{bg}$ and the foreground generator $G^{fg}$ respectively. To trigger this decomposition, we perturb a foreground image with the random affine transformation matrix $\mathcal{T}_\theta$ right before the composition of two image components. The association between the latent code $c$ and its image observation $\hat{x}$ is learnt by optimizing the information-theoretic regularization that is based on the contrastive loss defined for their feature representations in the embedding space of the discriminator. The inference of fine-grained class clustering is made based on the distances between images' semantic features $\{h_i(\bullet)\}_{i=0}^{N-1}$, that are depicted with dotted lines, and the set of cluster centroids $\{l_i(\blacklozenge)\}_{i=0}^{Y-1}$ which are the fixed number of embedded latent codes $c$.

## 3.1 PRELIMINARIES

Our proposed method is built on FineGAN (Singh et al., 2019), which is in turn based on InfoGAN (Chen et al., 2016). InfoGAN learns to associate the latent code $c$ with its observation $\hat{x}$ by inducing the model to maximize the mutual information of the two, $\mathrm{I}(c, \hat{x})$. The latent code $c$ can take various form depending on the prior knowledge of the factor that we want to infer, and is set to follow the uniform categorical distribution when our purpose is to make the categorical inference on given dataset. FineGAN learns three such latent codes for hierarchical image generation, and each code is respectively used for background, super-class object, and sub-class object image synthesis. To facilitate this decomposition, FineGAN employs multiple pairs of generator and discriminator and trains an auxiliary background classifier using object bounding box annotations. It further demonstrates that the latent codes for object image synthesis can be also utilized for clustering an image dataset according to their fine-grained classes.

Our method differs from FineGAN in that, i) it employs the discriminator $D$ to infer clusters without requiring additional training of classifiers, and ii) it learns only one latent code $c$ that corresponds to the fine-grained class of a foreground object. The separate random noise $z$ is kept, that is another input of the generator, to model variations occurring in a background region. The noise value $z \in \mathbb{R}^{d^z}$ is sampled from the normal distribution $\mathcal{N}(0, I)$, and the latent code $c \in \mathbb{R}^Y$ is an 1-hot vector where the index $k$ that makes $c_k = 1$ is sampled from the uniform categorical distribution, $\mathcal{U}(0, Y-1)$. Specifically, the background generator $G^{bg}$ synthesizes a background image $\hat{x}^{bg} \in \mathbb{R}^{3 \times H \times W}$ solely from the random noise $z$, and the foreground generator $G^{fg}$ synthesizes a foreground mask $\hat{m} \in \mathbb{R}^{1 \times H \times W}$ and an object image $\hat{t} \in \mathbb{R}^{3 \times H \times W}$ using both $z$ and $c$. To model foreground variations, we convert an 1-hot latent code $c$ to a variable $c' \in \mathbb{R}^{d^c}$ whose value is sampled from the gaussian distribution $N(\mu_c, \sigma_c)$ where the mean $\mu_c$ and diagonal covariance matrix $\sigma_c$ are computed according to the original code $c$ (Zhang et al., 2017). The final image $\hat{x} \in \mathbb{R}^{3 \times H \times W}$ is the composition of generated image components summed by the hadamard product, as described in Fig. 2.

To achieve a fully unsupervised method, we leverage the scene decomposition method proposed by PerturbGAN (Bielski & Favaro, 2019). In specific, scene decomposition is triggered by perturbing foreground components $\hat{m}$ and $\hat{t}$ with the random affine transformation matrix $\mathcal{T}_\theta$ right before the final image composition. The parameters $\theta$ of the random matrix $\mathcal{T}_\theta$ include a rotation angle, scaling factor and translation distance, and they are all randomly sampled from the uniform distributions with predefined value ranges.

In summation, the image generation process is described as below:

$$c' \sim N(\mu_c, \sigma_c), \quad \hat{x}^{bg}, \hat{m}, \hat{t} = G(z, c'),$$
$$\hat{m}', \hat{t}' = \mathcal{T}_\theta(\hat{m}, \hat{t}), \quad \hat{x} = \hat{x}^{bg} \odot (1 - \hat{m}') + \hat{t}' \odot \hat{m}'. \tag{1}$$

## 3.2 Contrastive Fine-grained Class Clustering

We assume that data would be well clustered when i) they form explicitly discernable cluster boundaries in the embedding space $\mathcal{H}$, and ii) each cluster centroid $l_y \in \mathbb{R}^{d^h}$ condenses distinct semantic features. This is the exact space that the discriminator of InfoGAN aims to approach when the latent code $c$ is sampled from the uniform categorical distribution, $\mathcal{U}(0, Y-1)$. However, since it jointly optimizes the adversarial loss, the model has a possibility of falling into the mode collapse phenomenon where the generator $G$ covers only a few confident modes (classes) to easily deceive the discriminator $D$. This is the reason why InfoGAN lacks the ability of inferring clusters on real image datasets. To handle this problem, we propose a new form of an auxiliary probability $Q(c|x)$ that represents the mutual information between the latent code $c$ and image observation $\hat{x}$.

Let us now describe the objective functions with mathematical definitions. The discriminator $D$ aims to learn an adversarial feature $r \in \mathbb{R}$ for the image authenticity discrimination, and a semantic feature $h \in \mathbb{R}^{d^h}$ for optimizing the information-theoretic regularization. The features are encoded from separate branches $\psi_x^r$ and $\psi_x^h$, which were split at the end of the base encoder of the discriminator $D_{base}$. The adversarial feature $r$ is learnt with the hinge loss as represented in the equations below:

$$\mathcal{L}_D^{adv} = \mathbb{E}_{x \sim X, \hat{x} \sim G(z,c)}\big[\min(0, 1-r) + \min(0, 1+\hat{r})\big], \qquad \mathcal{L}_G^{adv} = \mathbb{E}_{\hat{x} \sim G(z,c)}\big[-\hat{r}\big]$$
$$s.t. \qquad r = \psi_x^r(D_{base}(x)). \tag{2}$$

Meanwhile, the semantic feature $h$ can be considered as a logit in a classification model. In a supervised setting where ground-truth class labels are given, data mapped into an embedding space gradually form discernable cluster boundaries as the training proceeds, similar to the one in Fig. 1 (b). However, in an unsupervised setting, it is difficult to attain such space simply by maximizing the mutual information between the latent code $c$ and its observation $\hat{x}$ for the aforementioned reasons. Our solution is to formulate an auxiliary probability $Q(c|x)$ in the form of contrastive loss. We map the latent code $c$ onto the embedding space $\mathcal{H}$ with a simple linear layer $\psi_c$, and let it act as a cluster centroid ($l$) that pulls semantic features of images ($h$) that were generated from its specific value. By setting the relations with other latent codes as negative pairs, cluster centroids are set to be pushing each others and form distinguishable boundaries in the embedding space. Specifically, we expect the semantic feature $h$ to be mapped to the $k$-th cluster centroid, where $k$ is the index of its corresponding latent code $c$ that makes $c_k = 1$, while distances to other centroids $l_y$ are far enough to maximize the mutual information of the two. To enhance the robustness of the model and assist the scene decomposition learning, we additionally maximize the mutual information between the latent code $c$ and masked foreground images $\hat{x}^{fg}$. To sum up, the proposed information-theoretic objectives are defined as follows:

$$\mathcal{L}^{info} = \mathbb{E}_{\hat{x} \sim G(z,c)}\big[-\log \hat{q}_k\big], \qquad \mathcal{L}^{info^{fg}} = \mathbb{E}_{\hat{x} \sim G(z,c)}\big[-\log \hat{q}_k^{fg}\big]$$
$$s.t. \quad q_k = Q(c_k = 1|x) = \frac{\exp(\text{sim}(h, l_k)/\tau)}{\sum_{y=0}^{Y-1} \exp(\text{sim}(h, l_y)/\tau)}, \quad h = \psi_x^h(D_{base}(x)), \quad l = \psi_c(I_Y). \tag{3}$$

where $\text{sim}(a, b)$ is the cosine distance between vectors $a$ and $b$, $y$ is an index of a cluster, and $\tau$ is the temperature of the softmax function. $I_Y$ denotes the identity matrix with the size of $Y \times Y$.

**Additional Regularizations**  Following the prior works (Ji et al., 2019; Van Gansbeke et al., 2020), we adopt the overclustering strategy to help the model learn more expressive feature set. We also regularize the prediction on a real image $q \in \mathbb{R}^Y$ by minimizing its entropy $H(q)$ to promote each data to be mapped to only one cluster id with high confidence, along with the minimization of the KL divergence between batch-wisely averaged prediction $\bar{\mathbf{q}}$ and the uniform distribution $\mathbf{u}$, that is for avoiding a degenerated solution where only a few clusters are overly allocated. Furthermore, we optimize the contrastive loss (Chen et al., 2020) defined for real image features $h$ to assist the semantic feature learning. To summarize, the regularizations on real images are as follows:

$$\mathcal{L}^{img\_cont} = \mathbb{E}_{x \sim X}\big[-log \frac{\exp(\text{sim}(h, h')/\tau)}{\sum_{j=0}^{N-1} \exp(\text{sim}(h, h'_j)/\tau)}\big],$$
$$\mathcal{L}^{entropy} = \mathbb{E}_{x \sim X}[H(q)] + D_{\text{KL}}(\bar{\mathbf{q}}\|\mathbf{u}) \qquad s.t. \qquad \bar{\mathbf{q}} = \mathbb{E}_{x \sim X}[q]. \tag{4}$$

We also enforce a foreground mask $\hat{m}$ to be more like a hard mask and take up a reasonable portion of a scene by employing below regularization function.

$$
\begin{aligned}
\mathcal{L}^{mask} = \mathbb{E}_{\hat{x} \sim G(z,c)}[\, &\mathbb{E}_{hw}[-\hat{m} \, \log(\hat{m}) - (1 - \hat{m}) \, \log(1 - \hat{m})] \\
&+ \max(0, 0.1 - \mathbb{E}_{hw}[\hat{m}]) + \max(0, \mathbb{E}_{hw}[\hat{m}] - 0.9)\,].
\end{aligned}
\tag{5}
$$

In sum, C3-GAN is trained alternately optimizing the following objective functions for $D$ and $G$:

$$
\min_{D,G} \; \mathcal{L}_D^{adv} + \mathcal{L}_G^{adv} + \lambda_0 \mathcal{L}^{info} + \lambda_1 \mathcal{L}^{info^{fg}} + \lambda_2 \mathcal{L}^{img\_cont} + \lambda_3 \mathcal{L}^{entropy} + \lambda_4 \mathcal{L}^{mask}.
\tag{6}
$$

# 4 EXPERIMENTS

## 4.1 EXPERIMENTAL SETTING

**Datasets.** We tested our method on 4 datasets that consist of single object images. **i) CUB** (Wah et al., 2011): 5,994 training and 5,794 test images of 200 bird species. **ii) Stanford Cars** (Krause et al., 2013): 8,144 training and 8,041 test images of 196 car models. **iii) Stanford Dogs** (Khosla et al., 2011): 12,000 training and 8,580 test images of 120 dog species. **iv) Oxford Flower** (Nilsback & Zisserman, 2008): 2,040 training and 6,149 test images of 102 flower categories. Due to the small number of training images, all models for CUB and Oxford Flower datasets were trained with the entire dataset as the prior works did (Singh et al., 2019; Li et al., 2020; Benny & Wolf, 2020).

**Implementation Details.** The weights of the loss terms $(\lambda_0, \lambda_1, \lambda_2, \lambda_3, \lambda_4)$ are set as (5, 1, 1, 0.1, 1), and the temperature $\tau$ is set as 0.1. We utilized Adam optimizer of which learning rate is 0.0002 and values of momentum coefficients are (0.5, 0.999). The architectural specification of C3-GAN and the values of hyperparameters, such as the parameter ranges of the random affine transformation and the number of clusters $Y$ can be found in Appendix A.1 and Appendix A.2, respectively.

**Baselines.** We first compared C3-GAN with **IIC** (Ji et al., 2019) in order to emphasize the difference between the coarse-grained and fine-grained class clustering task, since it achieved decent performance with sobel filtered images where color and texture informations are discarded. We also experimented with simple baseline of **SimCLR+K-Means** (Chen et al., 2020) to claim that optimizing instance discrimination task is not enough for fine-grained feature learning. **SCAN** (Van Gansbeke et al., 2020), the work that shows a remarkable coarse-grained class clustering performance, is also compared to investigate whether the method of inducing consistent predictions for all nearest neighbors would be helpful for the fine-grained class clustering task as well. Finally, we compared with the methods that learn hierarchically fine-grained feature representations, such as **FineGAN** (Singh et al., 2019), **MixNMatch** (Li et al., 2020) and **OneGAN** (Benny & Wolf, 2020), including their base model **InfoGAN** (Chen et al., 2016), to claim the efficacy of the proposed formulation that is based on the contrastive loss. We mainly compared with the unsupervised version of them that is trained with pseudo-labeled bounding boxes which assign edges of real images as background region.

## 4.2 FINE-GRAINED CLASS CLUSTERING RESULTS

**Quantitative results.** We evaluated clustering performance with two metrics: Accuracy (Acc) and Normalized Mutual Information (NMI). The score of accuracy is calculated following the optimal mapping between cluster indices and real classes inferred by Hungarian algorithm (Kuhn & Yaw, 1955) for a given contingency table. We presented the results in Table 1. As it can be seen, C3-GAN outperforms other methods by remarkable margins in terms of Acc on all datasets. Our method presents better or comparable performance in terms of NMI scroes as well. The results of IIC underline that understanding only structural characteristic is not enough for the fine-grained class clustering task. From the results of SCAN (2nd rank) and MixNMatch (3rd rank), we can conjecture that both requiring local disentanglement of the embedding space and extracting foreground features via scene decomposition learning can be helpful for improving the fine-grained class clustering performance. However, it is difficult to determine whether the GAN-based methods or the SSL-based methods are better because both approaches have similar score ranges for all metrics and display globally entangled feature spaces as shown in Fig. 3 (a) and (b). Since C3-GAN resembles

|  | Acc ↑ | | | | NMI ↑ | | | |
|---|---|---|---|---|---|---|---|---|
|  | Bird | Car | Dog | Flower | Bird | Car | Dog | Flower |
| FineGAN[†] (Singh et al., 2019) | *12.6* | *7.8* | *7.9* | *-* | *0.40* | *0.35* | *0.23* | *-* |
| MixNMatch[†] (Li et al., 2020) | *13.6* | *7.9* | *8.9* | *-* | *0.42* | *0.36* | *0.32* | *-* |
| OneGAN[†] (Benny & Wolf, 2020) | *10.1* | *6.0* | *7.3* | *-* | *0.39* | *0.27* | *0.21* | *-* |
| *Fully Unsupervised Setting* | | | | | | | | |
| IIC (Ji et al., 2019) | 7.4 | 4.9 | 5.0 | 8.7 | 0.36 | 0.27 | 0.18 | 0.24 |
| SimCLR (Chen et al., 2020) +$k$-Means | 8.4 | 6.7 | 6.8 | 12.5 | 0.40 | 0.33 | 0.19 | 0.29 |
| InfoGAN (Chen et al., 2016) | 8.6 | 6.5 | 6.4 | 23.2 | 0.39 | 0.31 | 0.21 | 0.44 |
| FineGAN w/o labels | 6.9 | 6.8 | 6.0 | 8.1 | 0.37 | 0.33 | 0.22 | 0.24 |
| MixNMatch w/o labels | 10.2 | 7.3 | 10.3 | 39.0 | 0.41 | 0.34 | 0.30 | 0.57 |
| SCAN (Van Gansbeke et al., 2020) | 11.9 | 8.8 | 12.3 | 56.5 | 0.45 | 0.38 | 0.35 | **0.77** |
| C3-GAN (Ours) | **27.6** | **14.1** | **17.9** | **67.8** | **0.53** | **0.41** | **0.36** | 0.67 |

Table 1: Quantitative evaluation of clustering performance. [†] denotes that the values are reported ones in the original papers, and the rest scores are obtained by experimenting with the released codes of baselines on our set of evaluation datasets. Please note that the original methods of FineGAN, MixNMatch, and OneGAN utilize human-annotated labels.

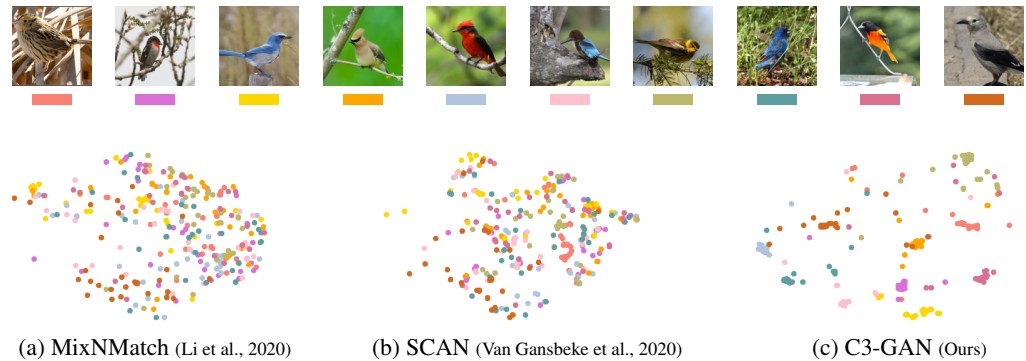

(a) MixNMatch (Li et al., 2020)       (b) SCAN (Van Gansbeke et al., 2020)       (c) C3-GAN (Ours)

Figure 3: Visualization of embedding spaces of MixNMatch, SCAN and C3-GAN that were trained on CUB dataset. To this end, we reduced the semantic features of images sampled from 10 classes into a 2-dimensional space through t-SNE. This result implies that only C3-GAN has learnt the embedding space where clusters are explicitly separable.

MixNMatch to some extent, its state-of-the-art scores can be partially explained by the fact that it learns decomposed foreground features, but the embedding space of C3-GAN visualized in Fig. 3 (c) implies that the performance of it has been further improved from our formulation of $Q(c|x)$ that enforces clusters to be distinctively distributed in the embedding space. We additionally presented the results of qualitative analysis on clustered images inferred by the top three methods in Appendix A.3, which better demonstrate the excellence of our method compared to prior works.

**Ablation Study.** We conducted an ablation study regarding three factors: i) Overclustering, ii) Foreground perturbation, and iii) The information-theoretic regularization based on the contrastive loss. We investigated clustering performance of C3-GAN on CUB dataset by removing each factor. The results can be seen in Table 2. The largest performance degradation was caused by changing the formulation of the information-theoretic regularization to a trivial softmax function. This result implies that clustering performance has a high correlation with the degree of separability of clusters, which was significantly improved by our method. Regarding the result of ii), we observed that C3-GAN fails to decompose a scene when the foreground perturbation was not implemented. This degenerated result suggests that the method of extracting only foreground features has a non-negligible amount of influence in clustering performance. Lastly, the result of i) implies that the overclustering strategy is also helpful for boosting performance since it allows a larger capacity for model to learn more expressive feature set. The additional analysis on hyperparameters can be found in Appendix A.2.

|  |  |  | Acc ↑ | NMI ↑ |
|---|---|---|---|---|
| C3-GAN (Ours) |  |  | **27.6** | **0.53** |
| i) | – | Overclustering | 22.7 | 0.50 |
| ii) | – | Foreground perturbation | 16.6 | 0.47 |
| iii) | – | The information-theoretic regularization based on the contrastive loss | 14.5 | 0.45 |

Table 2: Result of an ablation study. We observed that the largest performance degradation was caused by changing the formulation of the information-theoretic regularization to a trivial softmax function. The other two factors also have a non-negligible impact on performance.

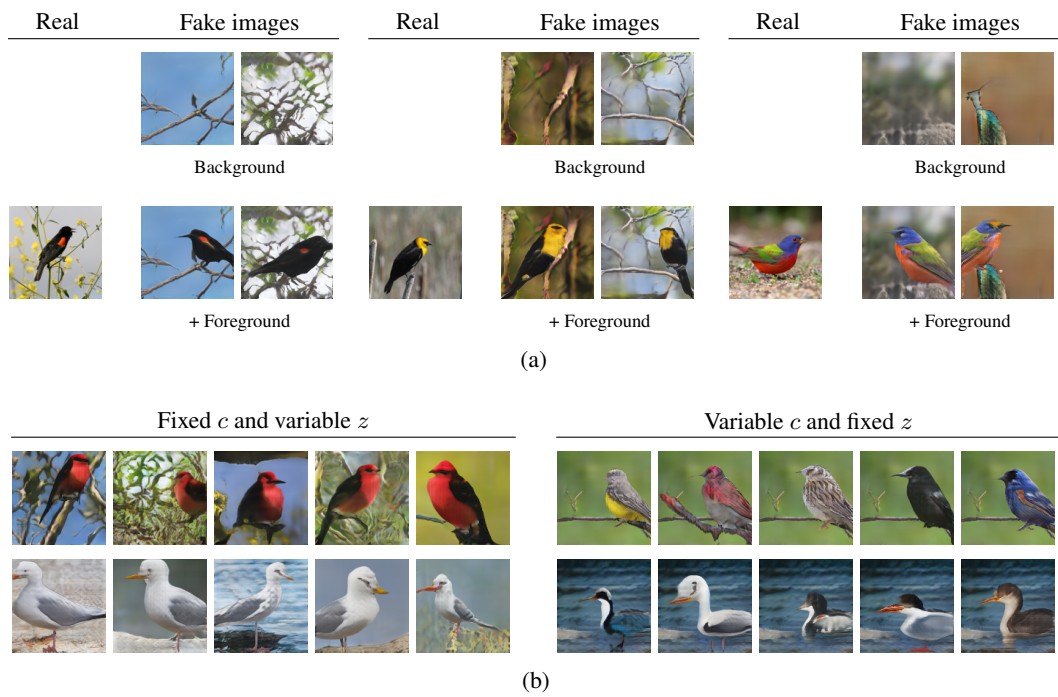

Figure 4: Qualitative analysis of the image generation performance. We present (a) images synthesized with the cluster indices of real images that were predicted by the discriminator, and (b) images synthesized by controlling values of the latent code $c$ and the random noise $z$.

## 4.3 IMAGE GENERATION RESULTS

**Qualitative Results.** For qualitative evaluation, we analyzed the results on CUB dataset. In Fig. 4 (a), we displayed synthesized images along with their decomposed scene elements. The images were generated with the cluster indices of real images that were predicted by the discriminator. Considering the consistency of foreground object features within the same class and the uniqueness of each class feature, we can assume that C3-GAN succeeded in learning the set of clusters in which each cluster represents its unique characteristic explicitly. Further, to investigate the roles of the two input variables, $c$ and $z$, we analyzed the change in the synthesized images when they were generated by fixing one of the two and diversifying the other, as shown in Fig. 4 (b). As it can be seen, if $c$ is fixed, all images depict the same bird species, but the pose of a bird and background vary depending on the value of $z$. Conversely, when generated under the condition of variable $c$ and fixed $z$, images of various bird species with the same pose and background were observed. The results on other datasets show the same trend. Please refer to Appendix A.4.

It is also notable that C3-GAN can effectively alleviate the mode collapse issue with the proposed method. Fig. 5 present that only C3-GAN reflects intra-class deviations of various factors such as color, shape, and layout for the given class, while other baselines produce images only with the layout variation. We conjecture that this performance was achieved because the learnt embedding space of the discriminator better represents the real data distribution, allowing the generator to get quality

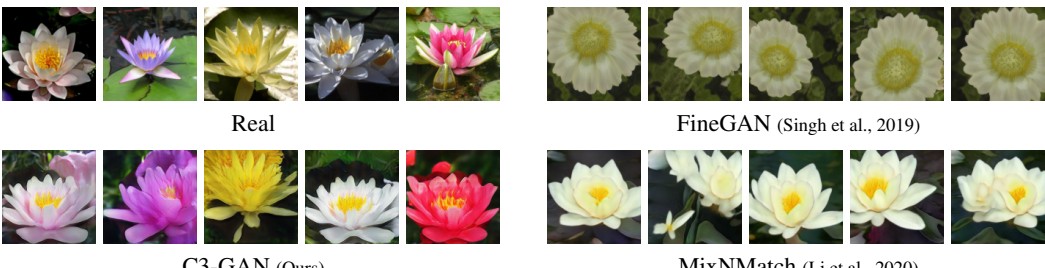

Figure 5: Images synthesized with the cluster index that corresponds to the class of real samples. Only C3-GAN generates images reflecting the *intra*-class color, shape, and layout variations of the real image set.

| | FID ↓ | | | | IS ↑ | | | | Reverse KL ↓ | | | |
|---|---|---|---|---|---|---|---|---|---|---|---|---|
| | Bird | Car | Dog | Flower | Bird | Car | Dog | Flower | Bird | Car | Dog | Flower |
| InfoGAN (Chen et al., 2016) | 35.71 | 70.91 | 59.07 | 78.37 | 8.20 | 3.25 | 6.91 | 13.39 | 0.56 | 0.73 | 0.40 | 0.42 |
| FineGAN w/o labels | 33.87 | 87.66 | 49.12 | 35.72 | 10.07 | **4.20** | 10.67 | 10.42 | 0.67 | 0.58 | 0.24 | 0.52 |
| MixNMatch w/o labels | 31.59 | 78.36 | 48.11 | **32.03** | 9.76 | 4.11 | **10.70** | **16.48** | 0.52 | **0.57** | 0.22 | 0.34 |
| C3-GAN (Ours) | **19.37** | **67.36** | **45.40** | 64.19 | **14.52** | 3.62 | 9.33 | 13.75 | **0.29** | **0.57** | **0.18** | **0.25** |

Table 3: Quantitative evaluation of image synthesis performance.

signals for the adversarial learning. Additionally, we found that only C3-GAN succeeded in the scene decomposition learning when annotations were not utilized. In fact, when FineGAN and MixNMatch were trained without object bounding box annotations, they fell into degenerate solutions where the entire image is synthesized from the background generators.

**Quantitative Results.** We also quantitatively evaluated the image synthesis performance based on the scores of Frechet Inception Distance (FID) and Inception Score (IS). We also considered the scores of reverse KL which measures the distance between the predicted class distribution of real images and synthesized images. We compared C3-GAN with GAN-based models that were trained without object bounding box annotations. The scores of IS and reverse KL were measured with Inception networks that were fine-tuned on each dataset using the method of Cui et al. (2018)[1]. The results are displayed in Table 3. C3-GAN presents the state-of-the-art or comparable performance in terms of all metrics. This result means that, in addition to the class clustering performance, C3-GAN has competitive performance in generating fine-grained object images.

## 5 CONCLUSION

In this study, we proposed a new GAN-based method, C3-GAN, for unsupervised fine-grained class clustering which is more challenging and less explored than a coarse-grained clustering task. To improve the fine-grained class clustering performance, we formulate the information-theoretic regularization based on the contrastive loss. Also, a scene decomposition-based approach is incorporated into the framework to enforce the model to learn features focusing on a foreground object. Extensive experiments show that our C3-GAN not only outperforms previous fine-grained clustering methods but also synthesizes fine-grained object images with comparable quality, while alleviating mode collapse that previous state-of-the-art GAN methods have been suffering from.

---

[1]The accuracy of Inception Networks fine-tuned on CUB/Stanford Cars/Stanford Dogs/Oxford Flower datasets are 0.87/0.47/0.86/0.94, respectively.

ETHICS STATEMENT

Remarkable advancement of generative models has played a crucial role as AI tools for text (Brown et al., 2020; Kim et al., 2021), image (Karras et al., 2020; Choi et al., 2020b), audio (Sisman et al., 2020), and multimodal content generation (Tian et al., 2021; Yan et al., 2021). However, as its side effect, many malicious applications have been reported as well, thus leading to severe societal problems, such as deepfake (Dolhansky et al., 2020), fake news (Lazer et al., 2018), and generation biased by training data (Gupta et al., 2021). Our work might be an extension of the harmful effects of generative models. On the contrary, generative models can be a solution to alleviate these side effects via data augmentation (Yang et al., 2019; Choi et al., 2020a). In particular, our method can contribute to alleviating data bias and fine-grained class imbalance. Therefore, many endless efforts are required to make these powerful generative models benefit humans.

ACKNOWLEDGMENTS

This work was experimented on the NAVER Smart Machine Learning (NSML) platform (Sung et al., 2017; Kim et al., 2018). We are especially grateful to Jun-Yan Zhu and NAVER AI Lab researchers for their constructive comments.

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

# A  APPENDIX

## A.1  NETWORK ARCHITECTURE

We present detailed description of the architecture of C3-GAN in Table 4 and Table 5. The colored dots (●, ●, ●) are for indicating that the outputs of earlier layers are used as the inputs of later layers. The training was done with 2 NVIDIA-V100 GPUs, and we optimized a model until the convergence of the FID score.

| Module | Layers | Input size | Output size |
|---|---|---|---|
| $G_{bg}$ | Linear, BN, GLU, UnSqueeze | 64 ($\mathbf{z}$) | $512{\times}4{\times}4$ |
| | [Up, Conv(3,1), BN, GLU]×5 | $512{\times}4{\times}4$ | $64{\times}128{\times}128$ |
| | [Conv(3,1), BN, GLU, Conv(3,1), BN]×3 | $64{\times}128{\times}128$ | $64{\times}128{\times}128$ |
| | Conv(3,1), Tanh | $64{\times}128{\times}128$ | $3{\times}128{\times}128$ ($\hat{\mathbf{x}}^{\mathbf{bg}}$) |
| $N(\mu_c, \sigma_c)$ | Linear, BN, GLU | $Y$ ($\mathbf{c}$) | 8 ● |
| $G_{fg}^{base}$ | Linear, BN, GLU, Reshape | 64 ($\mathbf{z}$) | $512{\times}4{\times}4$ |
| | [Up, Conv(3,1), BN, GLU]×5 | (512+8 ●)$\times4\times4$ | $16{\times}128{\times}128$ |
| | [Conv(3,1), BN, GLU, Conv(3,1), BN]×3 | $16{\times}128{\times}128$ | $16{\times}128{\times}128$ ● |
| $G_{fg}^{m}$ | Conv(3,1), BN, GLU | $16{\times}128{\times}128$ ● | $64{\times}128{\times}128$ |
| | Conv(3,1), Sigm | $64{\times}128{\times}128$ | $1{\times}128{\times}128$ ($\hat{\mathbf{m}}$) |
| $G_{fg}^{t}$ | Conv(3,1), BN, GLU | (16● + $Y$ ($\mathbf{c}$))$\times128\times128$ | $16{\times}128{\times}128$ |
| | [Conv(3,1), BN, GLU, Conv(3,1), BN]×2 | $16{\times}128{\times}128$ | $16{\times}128{\times}128$ |
| | Conv(3,1), BN, GLU | $16{\times}128{\times}128$ | $16{\times}128{\times}128$ |
| | Conv(3,1), Tanh | $16{\times}128{\times}128$ | $3{\times}128{\times}128$ ($\hat{\mathbf{t}}$) |

Table 4: Architectural description of the generator of C3-GAN.

| Module | Layers | Input size | Output size |
|---|---|---|---|
| $D_{base}$ | Conv(4,2), LReLU | $3{\times}128{\times}128$ ($\mathbf{x}$) | $64{\times}64{\times}64$ |
| | [Conv(4,2), BN, LReLU]×4 | $64{\times}64{\times}64$ | $512{\times}4{\times}4$ |
| | Conv(3,1), BN, LReLU | $512{\times}4{\times}4$ | $512{\times}4{\times}4$ ● |
| $\psi_x^r$ | Conv(4,4), Squeeze | $512{\times}4{\times}4$ ● | 1 ($\mathbf{r}$) |
| $\psi_x^h$ | Conv(3,1), BN, LReLU | $512{\times}4{\times}4$ ● | $512{\times}4{\times}4$ |
| | Conv(4,4), Squeeze | $512{\times}4{\times}4$ | 512 ($\mathbf{h}$) |
| $\psi_c$ | Linear | $Y$ ($\mathbf{c}$) | 512 ($\mathbf{l}$) |

Table 5: Architecture description of the discriminator of C3-GAN.

- Conv($a$,$b$) : 2D Convolution layer with kernel size $a$ and stride $b$
- GLU : Gated Linear Units

## A.2 ADDITIONAL ABLATION STUDY

**Random affine transformation.** We experimented with two types of purtubing policy, a weak and strong random affine transformation. The detailed value ranges of their parameters are presented in Table 6. The type of perturbation used for each dataset was determined by manually checking randomly perturbed real images. The strong perturbation was only used for CUB dataset, and the rest datasets used the weak perturbation.

| Criteria | Scale | Rotation | Translation | Dataset |
|---|---|---|---|---|
| Weak perturbation | (0.9, 1.1) | (-2, 2) | (-0.08, 0,08) | Stanford Cars, Stanford Dogs, Oxford Flower |
| Strong perturbation | (0.8, 1.5) | (-15, 15) | (-0.15, 0,15) | CUB |

Table 6: Details of random affine transformation.

**Overclustering.** We use four datasets for the evaluation, including CUB, Standford Cars, Stanford Dogs and Oxford Flower. As we described in 3.2, we employ the overclustering policy which is to set the number of clusters to be multiple times of the actual number of classes. To find the optimal setting, we compared the results of C3-GAN by setting the number of clusters as 1, 2, and 3 times the actual number of classes. The cluster numbers for each dataset were determined based on the results in Table 7. We found that the performance generally tends to improve as the number of clusters increases. This implies that overclustering is indeed helpful for the expressive feature learning. To further investigate the results when we are not aware of the actual number of classes, we additionally conducted all experiments by setting the number of clusters as 100 (underclustering) or 500 (overclustering). The results are represented in Table 8. For the underclustering setting, we only report the scores of NMI, since some classes have no matching cluster indices. Despite the less promising performance of the underclustering case, this result implies that our method can be applied to any datasets whose cluster size is not available, if we set a large enough number of clusters.

| | | Acc ↑ | NMI ↑ | FID ↓ |
|---|---|---|---|---|
| CUB | × 1 | 22.7 | 0.50 | 18.75 |
| | × 2 | **27.6** | **0.53** | 19.37 |
| | × 3 | 26.3 | 0.51 | **17.13** |
| Stanford Cars | × 1 | 8.3 | 0.33 | **64.55** |
| | × 2 | 11.5 | 0.39 | 66.65 |
| | × 3 | **14.1** | **0.41** | 67.36 |
| Stanford Dogs | × 1 | 11.8 | 0.30 | 51.37 |
| | × 2 | 15.8 | 0.35 | 54.82 |
| | × 3 | **17.9** | **0.36** | **45.40** |
| Oxford Flower | × 1 | 55.6 | **0.72** | 75.12 |
| | × 2 | 61.7 | 0.67 | 74.59 |
| | × 3 | **67.8** | 0.67 | **64.19** |

Table 7: Results of hyperparameter search for overclustering.

| | Acc ↑ | | | | NMI ↑ | | | |
|---|---|---|---|---|---|---|---|---|
| # of clusters | Bird | Car | Dog | Flower | Bird | Car | Dog | Flower |
| Actual number of classes × 3 (Overclustering) | 26.3 | 14.1 | 17.9 | 67.8 | 0.51 | 0.41 | 0.36 | 0.67 |
| 100 (Underclustering) | - | - | - | - | 0.38 | 0.25 | 0.32 | 0.75 |
| 500 (Overclustering) | 23.7 | 12.1 | 20.5 | 70.8 | 0.51 | 0.40 | 0.39 | 0.65 |

Table 8: Clustering performance when the number of clusters are arbitrarily set.

### A.3 QUALITATIVE ANALYSIS OF CLUSTERING RESULTS

We present the clustered images of CUB and Oxford Flowers datasets, along with the results of MixNMatch and SCAN in Figs. 6 and 7. From these results, we can assume that our method is better at clustering compared to the baseline methods. It is worth noting that even incorrectly assigned images look very similar with the given bird species for C3-GAN, while the results of other methods contain objects that are quite deviated from the condition. We could observe the similar trend for Oxford Flowers dataset, as it is shown in Fig. 7.

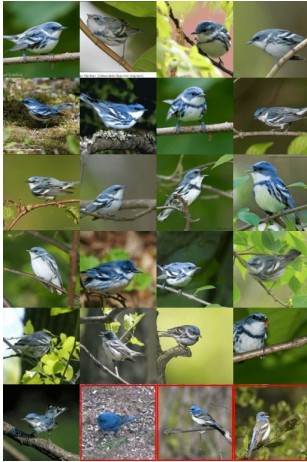 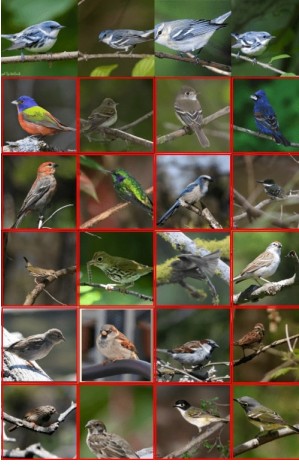 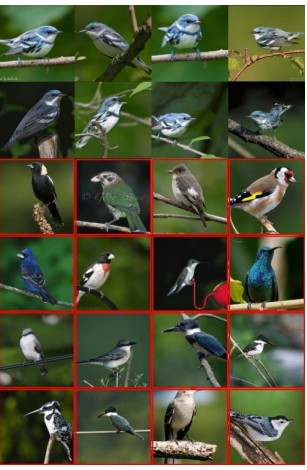

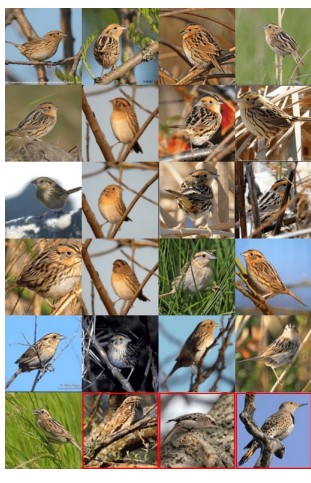 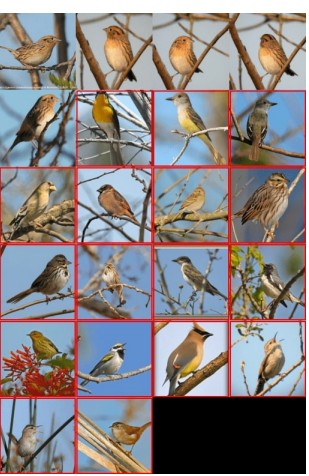

(a) C3-GAN (Ours)  (b) SCAN (Van Gansbeke et al., 2020)  (c) MixNMatch (Li et al., 2020)

Figure 6: Examples of clustered images of CUB dataset. The cluster indices for the first and the second sets are the ones mapped from the real classes *Cerulean Warbler* and *le Conte Sparrow* via Hungarian Algorithm. Red boxes denote incorrectly assigned images.

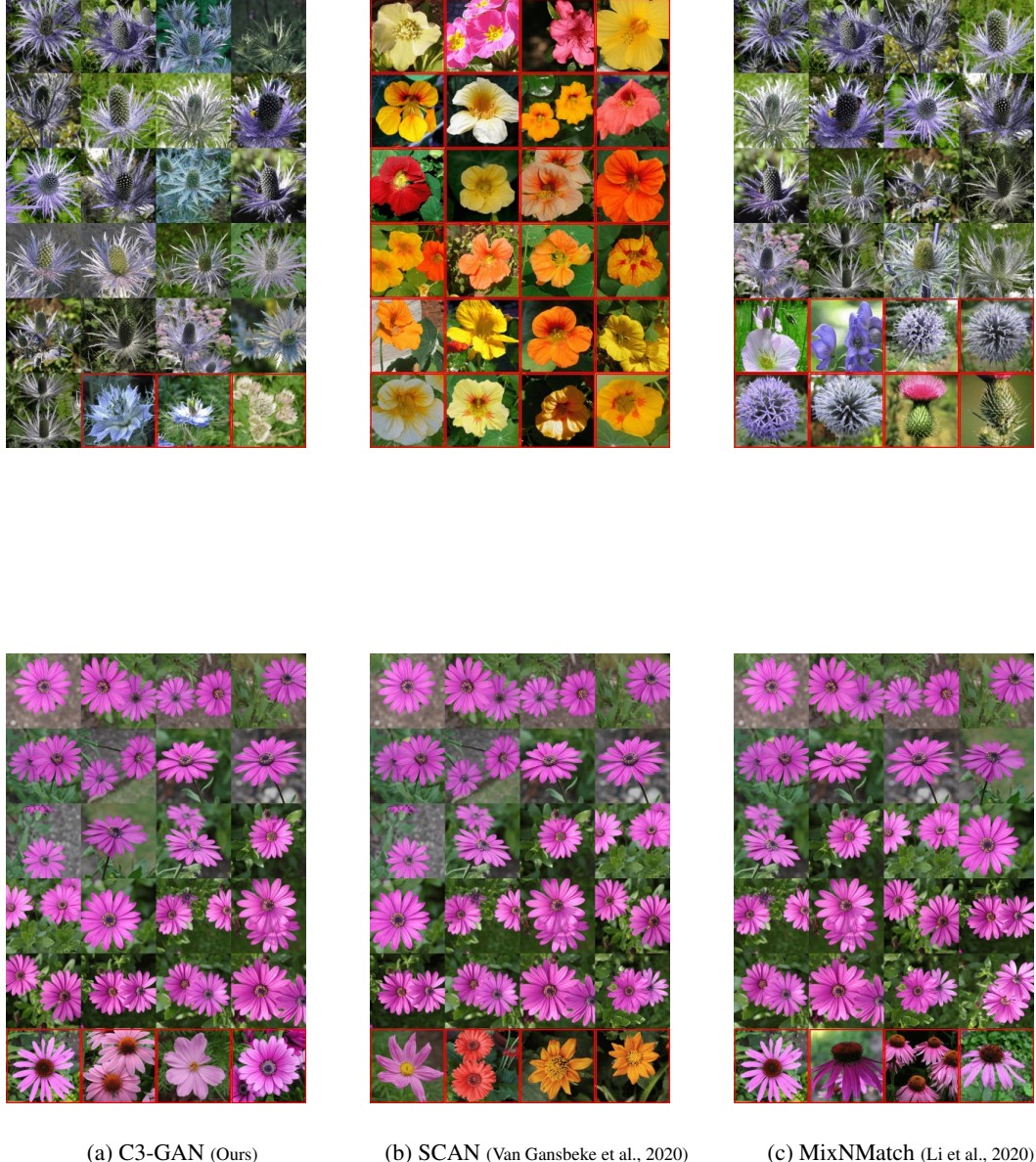

(a) C3-GAN (Ours)    (b) SCAN (Van Gansbeke et al., 2020)    (c) MixNMatch (Li et al., 2020)

Figure 7: Examples of clustered images of Flower dataset. The cluster indices for the first and the second sets are the ones mapped from the real classes *35* and *66* via Hungarian Algorithm. Red boxes denote incorrectly assigned images.

## A.4  ADDITIONAL IMAGE GENERATION RESULTS

### A.4.1  CUB

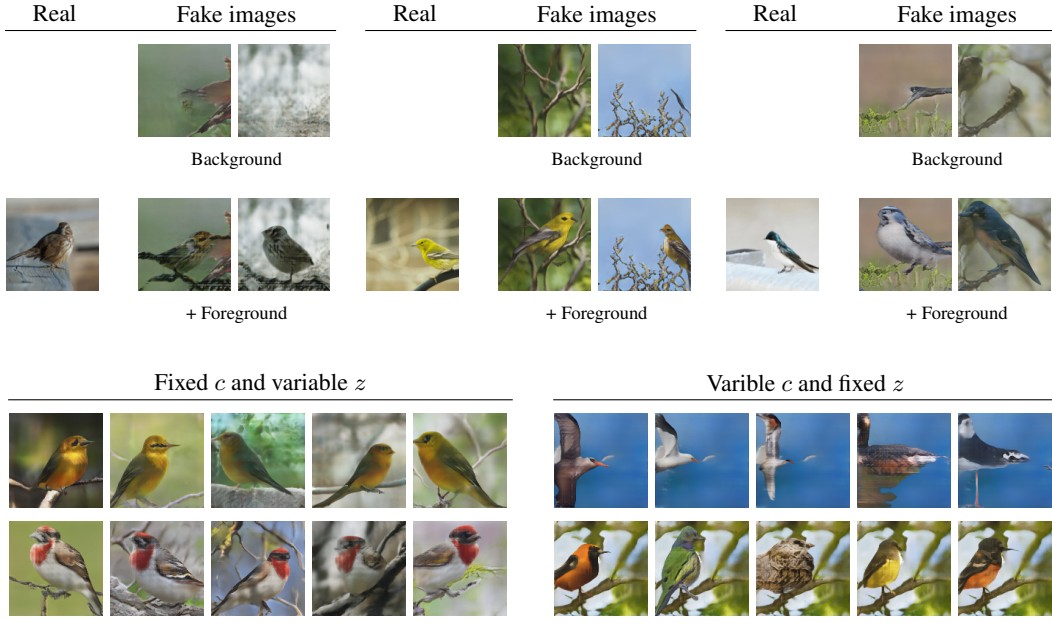

### A.4.2  STANFORD CARS

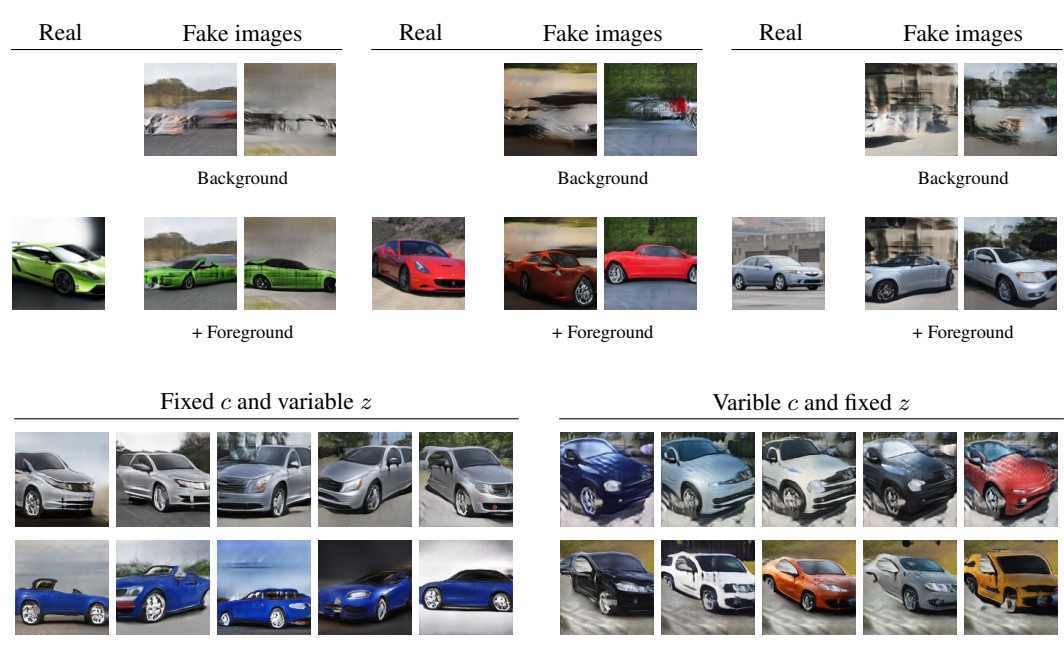

### A.4.3 STANFORD DOGS

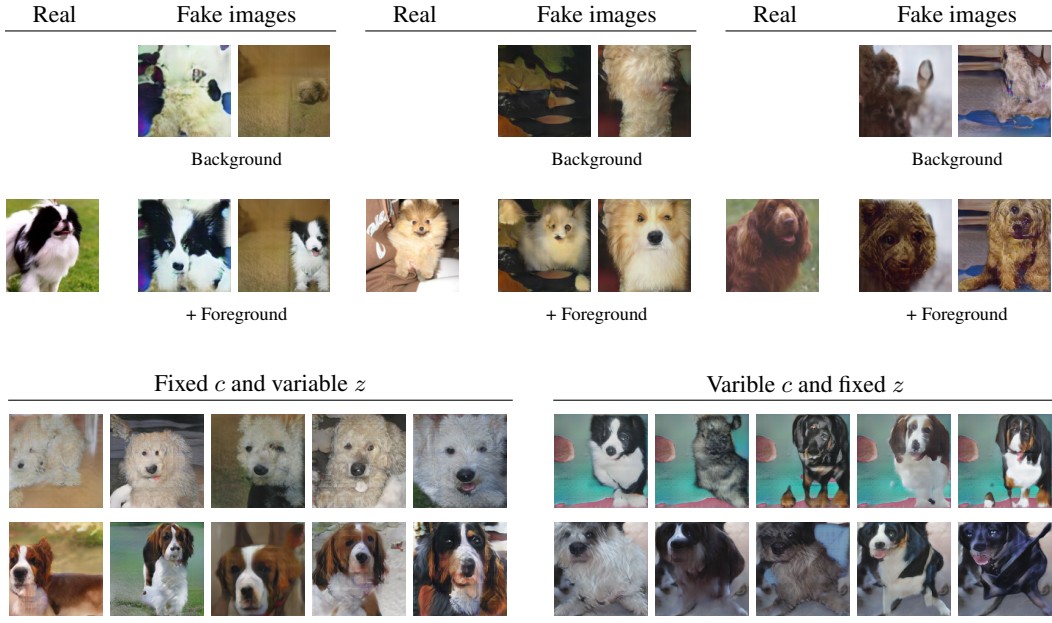

### A.4.4 OXFORD FLOWER

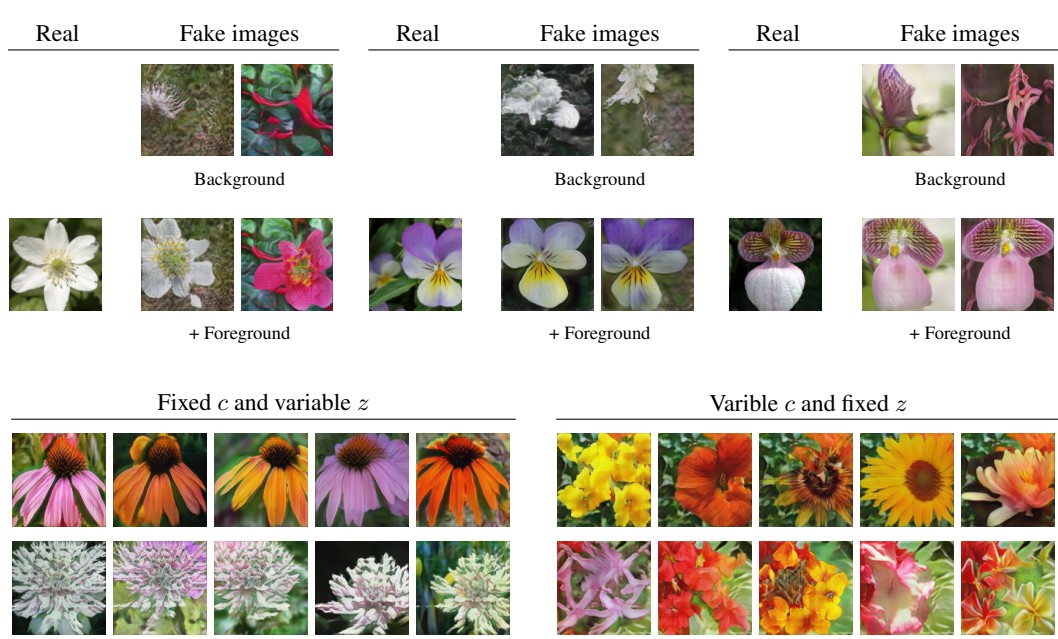

