# OpenReview forum: "Contrastive Fine-grained Class Clustering via Generative Adversarial Networks"
_ICLR.cc/2022/Conference — ICLR 2022 Spotlight_

### Official Review · Reviewer_5wFE · 2021-10-23

**Correctness:** 4
**Technical Novelty And Significance:** 3
**Empirical Novelty And Significance:** 3
**Recommendation:** 6
**Confidence:** 3

**Main Review:**

Strength:
	- The problem is known to be valid and challenging
	- The method idea, formulations, and figures are clear in general
	- The level of novelty is reasonable
	- The model is fully unsupervised
The authors provide an extensive evaluation and the results are impressive

Weaknesses:
	- The part of the contrastive loss is not totally clear. The authors should provide a better intuition of why the contrastive loss improves the feature representation. For example, how are image-latent pairs defined as positive?
	- The method focuses on learning cluster granularity for the object only, and not for the background.
	- It's unclear why the transformation matrix is used (other than the fact that it's part of PerturbGAN's pipeline)

A few comments on the text:
	- The phrase "coarse-grained images" is inaccurate, the "coarse-grained" adjective should refer to the clustering and not the images (in the intro).
	- The authors should share more details about the auxiliary distribution mentioned in the abstract and the intro.
	- Overall proofreading is required.
It would be great to add some of the model's notations to figure 2 (e.g. D_base, psi_r, psi_h)

**Summary Of The Paper:**

-- The authors propose C3-GAN, a method that learns "clustering-friendly" feature representation for fine-grained clustering (main goal), by learning features of cluster centroids (latent codes) using contrastive loss on the mutual information of image-latent code pairs.

-- The method should improve the GAN's performance in terms of image diversity.

-- The method is unsupervised and applicable for single-object images only.

-- The method is built upon FineGAN (and InfoGAN idea), after adding significant improvements such as removing the dependency on bounding-box labels, applying the mutual information in the embedding space, and directly learning cluster centroids.


**Summary Of The Review:**

I tend to vote for accepting this paper as I think it proposes a great approach and presents a convincing comparative performance.

---

> ### Author Response · Authors · 2021-11-15
> **Response to reviewer 5wFE**
>
> Thank you for the constructive suggestions. Here are our responses to your comments. And please check out the updated manuscript, since we revised the parts concerned with your comments marking with ***orange color***.
>
> > **Role of the random transformation matrix**
>
> $\to$ Despite its facility of fine-grained clustering, FineGAN [Singh et al. 2019] shows severely degenerated performances when bounding box annotations are not available. To achieve a fully unsupervised scene decomposition, we introduce the random transformation matrix, inspired by PerturbGAN [Bielski and Favaro, 2019]. We perturb the synthesized foreground images with the random transformation matrix, assuming that a realistic image will be composited even if the foreground image is slightly shifted under successful scene decomposition. Thanks to this mechanism, our model could successfully learn to decompose a scene in a complete unsupervised way. We would like to highlight that we achieved an effective way of successfully integrating two methods from FineGAN and PerturbGAN, for designing a “fully” unsupervised class clustering method, which is not trivial. We revised our manuscript (See Sec 1 in P1 and P2, Sec 3.1 in P4, and Sec 4.2 in P7).
>
> > **Definition of the positive image-latent pair & Role of the contrastive loss**
>
> $\to$ For generating foreground images, we randomly sample the latent code values from the uniform categorical distribution. Then this value is injected to the foreground generator in the form of a 1-hot vector. When this particular vector is mapped into the embedding space of the discriminator via a simple trainable linear matrix, it is considered as a cluster centroid. Our objective is to map the feature of the image which is synthesized with this latent value, close to this cluster centroid. So we define these two features as a positive pair, and set the relation with other cluster centroids as negative pairs. Optimizing the contrastive loss with this setting will render the discriminator to learn the embedding space where each centroid is distinctly separated which is optimal for clustering tasks. We more clarified these points in the revised manuscript (See Sec 1 in P2, Sec 3.2 in P5)
>
> > **Suggestions on editing**
>
> $\to$ Thank you for pointing out the parts that need to be strengthened. We agree with your opinion that the abstraction and the introduction can be improved. We substituted some sloppily used words with more detailed descriptions for reducing the possibility of confusion, and added more details on the methodological explanations. Thank you for your valuable feedback as a reader with a first impression! Please feel free to give us more comments if you find something still confusing in the revised version of the paper we just uploaded (See Figure 2 in P4 and Sec 3.2 in P5).
>
> - [Singh et al. 2019] Krishna Kumar Singh, Utkarsh Ojha, and Yong Jae Lee. Finegan: Unsupervised hierarchical disen-tanglement for fine-grained object generation and discovery. CVPR, 2019
>
> - [Bielski and Favaro, 2019] Emergence of object segmentation in perturbed generative models. NeurIPS, 2019.

---

> ### Author Response · Authors · 2021-11-29
> **Gentle remind**
>
> Dear Reviewer 5wFE,
>
> Thank you again for your constructive comments and suggestion.
> We gently remind you that final dicussion period will end soon.
> We have tried to alleviate your concerns, in particular, by more clarifying some less clear descriptions that you pointed out. Also, we revised our paper to be more clear.
> If you have any more questions and concerns, please let us know them.
>
> Best regards,
> Authors.

---

### Official Review · Reviewer_9p3B · 2021-10-27

**Correctness:** 4
**Technical Novelty And Significance:** 3
**Empirical Novelty And Significance:** 3
**Recommendation:** 8
**Confidence:** 3

**Main Review:**

Strengths

	The experiments are well executed, and clearly communicated.
	The quantitative results for clustering, and the qualitative figures in the supplementary material are both impressive, and they support the authors' claims
	The method has novel components which are provably attributed to these results.

Weaknesses

	The paper is well rounded, with all aspects detailed to completion. There are no obvious weaknesses

**Summary Of The Paper:**

The authors undertake the more difficult task of data clustering, based upon fine-grained features. They use contrastive learning for this, in conjunction with GAN losses. Their representation can be used in the downstream task of image generation, where they use their representations' strengths to show improved resilience to mode collapse, while displaying better intra-cluster variance.

**Summary Of The Review:**

The paper combines several techniques to achieve unsupervised fine-grained clustering of semantic classes. Their representations' high quality is able to drive GAN generation without mode collapse, thereby achieving great two-fold contributions.

---

> ### Author Response · Authors · 2021-11-15
> **Response to reviewer 9p3B**
>
> We are really grateful for your encouraging comments.
> Please feel free to add additional remarks regarding the content of the paper!

---

### Official Review · Reviewer_imaQ · 2021-11-02

**Correctness:** 3
**Technical Novelty And Significance:** 2
**Empirical Novelty And Significance:** 2
**Recommendation:** 6
**Confidence:** 4

**Main Review:**

[Strengths]
1. The method is straightforward and the paper is easy to follow.
2. The performance boost compared to SOTA for clustering seems to be very good (Table 1).
3. The ablation study covers major components in the pipeline so it is easy to understand their contributions.

[Weaknesses]
While the paper is focusing on fine-grained image synthesis/clustering, it is not specifically designed for fine-grained classes: the whole pipeline is pretty generic. This is unlike for example the FineGAN paper where they design separate stages for parent and child classes. Other than showing in the experiment that the proposed method just *works* in the fine-grained setting, I'd be interested in seeing more analysis on the advantage of the proposed pipeline in fine-grained setting vs. coarse-grained setting. For example, the paper could show the performance of parent and child classes in those datasets and compare against SOTA. If the proposed method has a similar performance as others in the coarse setting but a clear advantage in the fine-grained setting, that will be a very strong signal.

In addition, there are some parts in the paper that are unclear or confusing.
1. Section 3.2, second paragraph. "The discriminator D aims to learn adversarial features r' --> what does 'adversarial features' mean here? I checked the architecture it seems to be just the discriminator score (a scalar).
2. What is the effect of under-clustering where Y is much smaller?
3. The loss defined in Eq. 5 does not use the notion Q or q_k defined in Eq. 4. It is not clear from the text directly what their connection is. It should be written clearly L = -log q_k for example.
4. Table 1 last sentence: I think OneGAN is an unsupervised method.

[Other suggestions for editing]
1. The first sentence in the Abstract has redundant words: 'Unsupervisedly' and 'without ground truth annotation' are the same thing.
2. Figure 4 (a): the same cluster c is very confusing here because those birds are from different classes. I'd suggest simply dropping the symbol c.


**Summary Of The Paper:**

This paper studies the problem of fine-grained image clustering. Similar to recent work such as OneGAN and FineGAN, the paper proposes to use a GAN setup, called C-3GAN, where the fine-grained image synthesis and clustering are performed in the same end-to-end pipeline. The main contribution is to use a contrastive loss in maximizing the mutual information of the discriminator encoded features and class-centric features. There are also some modifications in the foreground and background synthesis mechanisms compared to prior work such as the FineGAN. Experiments are conducted on fine-grained datasets such as CUB-200-2011.

**Summary Of The Review:**

Overall, this paper proposes a straightforward pipeline to synthesize and cluster images in fine-grained classes. A contrastive loss is used in place of the regular softmax loss in the InfoGAN framework. The performance seems to be very solid compared to SOTA methods on both synthesis and clustering tasks. There are some unclear or confusing parts in the paper, but I think given the simplicity and good performance of the method, it might be worth being seen by the community to inspire similar works.

---

> ### Author Response · Authors · 2021-11-15
> **Response to reviewer imaQ (Cont.)**
>
> > **Suggestions on editing**
>
> $\to$ Following your suggestions, we revised the text and figures by adding details or eliminating some sloppily used words, and uploaded the revised version of the paper. They are marked with the blue colored text. We especially adjusted Figure 2. to represent all the outputs of the discriminator. The suggestions were really helpful for clarifying the content. Thank you!

---

> ### Author Response · Authors · 2021-11-15
> **Response to reviewer imaQ**
>
> Thank you for your constructive feedback. Here are our responses to your comments.
> And please check out the revised manuscript, since we clarified the parts concerned with your comments marking with ***blue color***.
>
>
> > **Results on coarse-grained clustering**
>
> $\to$ Thanks for suggesting an interesting experimental setup. We agree that the result of the experiments you suggested are worth analyzing.
>
> For validating the hierarchical clustering performance on both coarse-grained and fine-grained classes, we merged four fine-grained datasets used in the main experiments into the one dataset. We report the clustering performance on 4 coarse-grained classes (bird/car/dog/flower) and 618 fine-grained classes (200/196/120/102 classes for each object). We set the overclustering size to 1854. In the table below, we compare the results with MixNMatch, which is one of the SOTA GAN-based methods aiming to find the feature hierarchy.
>
> | Methods | Coarse-grained clustering acc | Fine-grained clustering acc | NMI | FID |
> | :---: | :---: |  :---: | :---: | :---: |
> | MixNMatch | 83.8 | 10.2 | 0.43 | 70.78 |
> | C3-GAN (ours) | 79.7 | 18.6 | 0.44 | 57.67 |
>
> The results show that C3-GAN outperforms MixNMatch in terms of fine-grained clustering accuracy and fid score, while demonstrating comparable performance in terms of coarse-grained clustering accuracy and nmi score. We think it is hard to further widen the gap with the current method since there are higher variances in both background and foreground characteristics in the merged dataset, which renders the scene decomposition learning more difficult. For better performances, therefore, it seems that a stronger backbone of GAN, such as BigGAN [Brock et al. 2019], and additional regularizations with more sophisticated design on hierarchical characteristics of latent codes might be required. We leave this as a future direction.
>
> > **Random number of clusters**
>
> $\to$ Thank you for pointing out this issue. We set the number of clusters as multiples of the number of ground-truth classes ( $C$ ) for evaluating accuracy scores by applying the Hungarian algorithm. In this setting, we replicate the contingency table of size $C \times (C \times n)$, to the matrix of size $(C \times n) \times (C \times n)$, where $n$ is a multiplier, so that each class is allocated with $n$ clusters at one iteration. We agree with your opinion that the setting where no prior knowledge is considered should also be investigated. Hence, we additionally conducted all experiments by setting the number of clusters as 100 (underclustering) or 500 (overclustering). For the underclustering setting, we only report the scores of NMI since some classes have no matching cluster indices. For evaluating the overclustering case, we applied the Hungarian algorithm multiple times with the original contingency table, by eliminating columns mapped in the prior turn, and the number of allocated clusters are varied for each class. The results are summarized in the table below:
>
> | Acc | Bird | Car | Dog | Flower |
> | :---: | :---: | :---: | :---: | :---: |
> | $n$=3 | 26.3 | 14.1 | 17.9 | 67.8 |
> | 500 (over) | 23.7 | 12.1 | 20.5 | 70.8 |
>
> | NMI | Bird | Car | Dog | Flower |
> | :---: | :---: | :---: | :---: | :---: |
> | $n$=3 | 0.51 | 0.41 | 0.36 | 0.67 |
> | 100 (under) | 0.38 | 0.25 | 0.32 | 0.75 |
> | 500 (over) | 0.51 | 0.40 | 0.39 | 0.65 |
>
> The result of the overclustering case, which is the same as our policy, has not deteriorated much, while the result of the underclustering setting presents lower performance in terms of NMI. This is coincident with the result in Appendix A.2, which shows that there is a correlation between the clustering performance and the number of clusters. Despite the less promising results in the underclustering setting, the result in the table above implies that our method can be applied to any fine-grained object datasets where the numbers of object classes are unknown, if we set a large enough number of clusters. We included this result in Appendix A.2 in the revised manuscript.
>
> > **Comment regarding OneGAN**
>
> $\to$ We referred to a work as a supervised method if they utilize any types of human annotated labels such as bounding boxes. As far as we know, OneGAN [Benny and Wolf. 2020] also utilizes bounding box annotations for training an auxiliary background classifier by generating clean background patches. We are afraid that we couldn’t add the results of the unsupervised version of OneGAN since its code implementation has not been released.
>
> - [Brock et al. 2019] Andrew Brock, Jeff Donahue, Karen Simonyan. Large Scale GAN Training for High Fidelity Natural Image Synthesis. ICLR 2019.
>
> - [Benny and Wolf. 2020] Yaniv Benny and Lior Wolf.  Onegan: Simultaneous unsupervised learning of conditional image generation, foreground segmentation, and fine-grained clustering. ECCV, 2020.

---

> > ### Comment · Reviewer_imaQ · 2021-11-29
> > **The additional analysis and revision looks good**
> >
> > I like the additional analysis results on coarse-vs-finegrained performance and the effects of underclustering. They add more insights into a better understanding of the proposed framework. I also looked at the revised draft and it addresses most of my concerns in the original reviews. Therefore I'd keep my rating and vote for acceptance.

---

### Official Review · Reviewer_odf8 · 2021-11-03

**Correctness:** 4
**Technical Novelty And Significance:** 3
**Empirical Novelty And Significance:** 2
**Recommendation:** 8
**Confidence:** 3

**Main Review:**

*Strengths*
+ Information theory based regularization with contrastive loss learns the features of the cluster centroids. The proposed method is well adapted for this purpose using InfoGAN and PerturbGAN.
+ The proposed method effectively avoids mode collapse while training generators, and are able to generate, with good control, various objects with varying background (while keeping foreground fixed), and varying foreground objects (while keeping background fixed) better than related methods.
+ The paper demonstrates that overclustering is a viable approach to hyper-parameter optimization that can achieve better fine-grained clustering due to learning dense features.

*Weaknesses*
+ The approach is heavily inspired by FineGAN and PerturbGAN.
+ In overclustering, it is not clear what would happen if the chosen cluster size is such that it is not a multiple of no. of classes? We assume that the data does not have labels in unsupervised feature learning, therefore it would be interesting to see if the cluster size is not a multiple.

*Grammatical/Typographical errors*
+ Page 1, last line: `...we formulate the auxiliary probability...`
+ Fig. 2 caption: `C-3GAN`.
+ Page 5, first line: `...In mathematical, ...`

**Summary Of The Paper:**

*The fine-grained class clustering is more challenging than coarse-grained due to lower sample representation and large scale- and color- variations between the fine-grained classes. The main goal of the proposed approach is to learn stronger representations in an unsupervised fashion. To this end, the paper proposes C3-GAN which uses the ability of InfoGAN with contrastive learning to learn feature representations that maximize the mutual information between the latent code and its corresponding observation. The proposed approach is able to achieve best performance in comparison to the related works.*

*The results demonstrated quantitatively and qualitatively on 4 different datasets along with ablation study validate the proposed approach. The method is promising since it is unsupervised way of learning cluster centroid for unlabeled data.*

**Summary Of The Review:**

*The paper presents a method that is a hybrid of 2 major previous state-of-art methods. Although heavily inspired, the paper proposes solutions to reduce the drawbacks of the previous approaches - which is the highlight of the contribution. Additionally, the biggest take-away is that the method is unsupervised. Given the breadth of the experiments to validate each of the proposed solutions, and substantial ablation experiments to justify each proposal, the paper overall is a good contribution.*

---

> ### Author Response · Authors · 2021-11-15
> **Response to reviewer odf8**
>
> Thank you for the constructive suggestions. Here are our responses to your comments.
>
> > **Random number of clusters**
>
> $\to$ Thank you for pointing out this issue. We set the number of clusters as multiples of the number of ground-truth classes ( $C$ ) for evaluating accuracy scores by applying the Hungarian algorithm. In this setting, we replicate the contingency table of size $C \times (C \times n)$, to the matrix of size $(C \times n) \times (C \times n)$, where $n$ is a multiplier, so that each class is allocated with $n$ clusters at one iteration. We agree with your opinion that the setting where no prior knowledge is considered should also be investigated. Hence, we additionally conducted all experiments by setting the number of clusters as 100 (underclustering) or 500 (overclustering). For the underclustering setting, we only report the scores of NMI since some classes have no matching cluster indices. For evaluating the overclustering case, we applied the Hungarian algorithm multiple times with the original contingency table, by eliminating columns mapped in the prior turn, and the number of allocated clusters are varied for each class. The results are summarized in the table below:
>
> | acc | bird | car | dog | flower |
> | :---: | :---: | :---: | :---: | :---: |
> | $n$=3 | 26.3 | 14.1 | 17.9 | 67.8 |
> | 500 (over) | 23.7 | 12.1 | 20.5 | 70.8 |
>
> | nmi | bird | car | dog | flower |
> | :---: | :---: | :---: | :---: | :---: |
> | $n$=3 | 0.51 | 0.41 | 0.36 | 0.67 |
> | 100 (under) | 0.38 | 0.25 | 0.32 | 0.75 |
> | 500 (over) | 0.51 | 0.40 | 0.39 | 0.65 |
>
> The result of the overclustering case, which is the same as our policy, has not deteriorated much, while the result of the underclustering setting presents lower performance in terms of NMI. This is coincident with the result in Appendix A.2, which shows that there is a correlation between the clustering performance and the number of clusters. Despite the less promising results in the underclustering setting, the result in the table above implies that our method can be applied to any fine-grained object datasets where the numbers of object classes are unknown, if we set a large enough number of clusters. We included this result in Appendix A.2 in the revised manuscript.
>
> > **Relation to FineGAN and PerturbGAN**
>
> $\to$ We agree with your opinion that our novelty seems incremental, since it is largely inspired by FineGAN [Singh et al. 2019] and PerturbGAN [Bielski and Favaro, 2019]. As you mentioned as well, however, our novelty can be found in the way we integrate these two to design a “completely” unsupervised clustering method. Both of these cannot solve the class clustering task in an unsupervised way, since FineGAN requires bounding box annotations and PerturbGAN does not aim to learn image feature representations. Further, we claim that designing the objective function based on the contrastive loss between the image feature and the latent feature adds the degree of novelty of our work. We clarify this point in the revised manuscript.
>
> > **Typos**
>
> $\to$ Thank you for your suggestions. We fixed the typos in the revised manuscript.
>
> - [Singh et al. 2019] Krishna Kumar Singh, Utkarsh Ojha, and Yong Jae Lee. Finegan: Unsupervised hierarchical disen-tanglement for fine-grained object generation and discovery. CVPR, 2019
>
> - [Bielski and Favaro, 2019] Emergence of object segmentation in perturbed generative models. NeurIPS, 2019.

---

> > ### Comment · Reviewer_odf8 · 2021-11-30
> > **Responses are satisfactory**
> >
> > Dear Authors,
> > Thanks for posting the additional experiments with the under- and over-clustering. Now, it is a bit more clear. What is interesting for me is the observation that optimal setting seems to be closer to 3 $\times n$. Add this to the additional experiments with random number of classes, we see that it might be a bit tricky to find the optimal number.
> >
> > Regardless, the paper seems to be in a better shape and I would increase my support by $+1$.

---

### Author Response · Authors · 2021-11-15
**Dear all reviewers,**

We appreciate all reviewers for their constructive comments and encouraging remark in terms of our challenging task (all reviewers), the simplicity and effectiveness of our method (all reviewers), intensive experiments and promising results (all reviewers), clear writing (imaQ, 9p3B), and moderate novelty (5wFE, 9p3B). Some reviewers gave concerns and questions: more analysis on overclustering effects (odf8, 5wFE), more clear description on our method (5wFE), and the efficacy on cluster hierarchy case (imaQ). We addressed the raised concerns in response for each review comment in detail. Also, we uploaded the revised manuscript reflecting reviewers’ comments (in colored fonts).

---

### Decision · Program_Chairs · 2022-01-20

**Decision:**

Accept (Spotlight)

**Comment:**

All the reviewers liked the paper. The proposed method contains novel ideas of learning feature representation to maixmize the mutral informatio nbetween the latent code and its corresponding observation for fine-grained class clustering. The model seems to successfully avoid mode collapse while training generators and able to generate various object (foregrounds) with varying backgrounds. The foreground and background control ability is an outstanding feature of the paper. Please incorporate the comments of the reviewers in the final version.

BTW, the real score of this paper should be 7.0 as Reviewer 5wFE commented that he/she would raise the score from 5 to 6 but at the time of this meta review, ths core was not raised. So the final score of the paper should be 8/8/6/6.